# Examining Factors Associated with Attrition, Strategies for Retention Among Undergraduate Nursing Students, and Identified Research Gaps: A Scoping Review

**DOI:** 10.3390/nursrep15060182

**Published:** 2025-05-22

**Authors:** Rohangez Lida Sheikoleslami, Daisy Michelle Princeton, Linda Iren Mihaila Hansen, Sezer Kisa, Alka Rani Goyal

**Affiliations:** 1Department of Nursing and Health Promotion, Faculty of Health Sciences, Oslo Metropolitan University, 0130 Oslo, Norway; rpshe@oslomet.no (R.L.S.); dapri@oslomet.no (D.M.P.); alka@oslomet.no (A.R.G.); 2Department of Health and Nursing Sciences, Faculty of Health and Sport Sciences, University of Agder, 4604 Kristiansand, Norway; linda.hansen@uia.no

**Keywords:** nursing education, student attrition, student dropout, retention strategies, social support

## Abstract

**Background/Objectives**: High-quality healthcare delivery relies on a on a sustainable nursing workforce. However, rising attrition rates and declining enrollment in nursing programs pose a significant challenge. A comprehensive synthesis of these factors for student attrition alongside effective retention strategies is needed to guide interventions. The aim of this scoping review is to map and synthesize existing evidence on the factors contributing to attrition among bachelor’s nursing students and to identify strategies that have been implemented or proposed to improve student retention in undergraduate nursing programs. **Methods:** Following the Preferred Reporting Items for Systematic Reviews and Meta-Analyses (PRISMA 2020) Checklist and Joanna Briggs Institute (JBI) guidelines, a systematic search was conducted in the following databases: MEDLINE/PubMed, Embase, Web of Science, PsycInfo, CINAHL, and Ovid. This review included peer-reviewed, English-language empirical studies (2010–December 2024) on attrition, dropout, or retention among bachelor-level nursing students, excluding non-nursing, non-bachelor programs, and unpublished studies or studies without primary data. A structured content analysis approach was used to synthesize findings from both qualitative and quantitative studies. **Results:** After screening titles, abstracts, and full texts, 19 articles were found eligible for inclusion. Analyses of the included studies revealed four key themes contributing to nursing student attrition: academic factors, institutional and social support, personal factors, and economic challenges. Retention strategies were categorized into two overarching themes: academic and non-academic approaches. **Conclusions:** Bachelor’s nursing programs should adopt retention strategies that enhance institutional and social support to reduce attrition. Strengthening supportive environments alongside curricular reform is key to building a resilient nursing workforce and ensuring quality care.

## 1. Introduction

Nursing education plays a critical role in preparing a competent workforce to meet global healthcare demands and to deliver high-quality care [1]. However, attrition in bachelor nursing programs remains a persistent and concerning challenge worldwide, with reported rates ranging from 10% to over 30%, depending on the country, program structure, and student demographics. Attrition rates vary across countries, with reports indicating 15–20% in Norway [2], 9% in Finland, 20% in England, 33% in Italy, and between 10% and 50% in Australia and Canada [3,4], as well as 20% in the United States [5,6]. High attrition rates not only impact individual students through lost educational opportunities, financial burdens, and emotional distress but also have broader consequences for the healthcare system, contributing to ongoing nursing shortages, increased pressure on remaining staff, and diminished quality of patient care.

The declining enrollment in, and rising attrition from, Bachelor of Nursing programs further exacerbate the shortage, leading to increased workloads, reduced job satisfaction, and challenges in maintaining patient safety and care quality [4,7,8,9]. Healthcare professionals in many countries face barriers that hinder the quality of patient care, patient safety, and patient satisfaction—challenges that became even more apparent following the COVID-19 pandemic. Reference [3] estimates a nine million global nursing shortage by 2030. Student attrition, also called dropout, is a significant contributor to this shortage [10], and the future of the nursing profession depends heavily on educational institutions’ ability to retain students [4]. Therefore, addressing the shortage requires increasing enrollment and improving student retention, making it imperative to understand the factors contributing to attrition and develop strategies to mitigate them.

Attrition in nursing education, which refers to students who leave or discontinue their studies before completing the program, is a multifactorial issue. Contributing factors vary widely and include academic challenges, lack of institutional support, financial difficulties, mental health concerns, and limited social integration [11,12]. These factors differ between first year and final-year students, with early-stage attrition commonly linked to unmet expectations and transition-related stress, while later-stage attrition is often influenced by clinical training pressures and personal circumstances. The factors are often interrelated and may vary depending on the stage of the nursing program. Various strategies have been proposed to improve retention, such as mentorship programs, academic support services, and financial assistance. However, the effectiveness of these interventions varies, and there is a lack of comprehensive synthesis of the evidence.

Attrition and retention are often treated as opposing concepts, even though the factors leading to attrition may differ from those that encourage persistence. Another challenge is the inconsistency in how attrition is measured, with variations in definitions and methodologies making cross-study comparisons difficult. Furthermore, while interventions such as academic support, mentoring, and financial aid have been explored, there is limited research on how institutions can integrate these strategies into cohesive retention models. Filling these knowledge gaps is essential for developing targeted and effective interventions that enhance student success and strengthen the nursing workforce. Given the complexity and multifaceted nature of attrition among nursing students, a scoping review is warranted to map the existing literature, identify the factors contributing to attrition in bachelor nursing education, and explore the strategies used to enhance student retention—organized thematically into academic, institutional, social, personal, and economic domains. This approach will provide a comprehensive picture of the current state of knowledge, highlight gaps in the literature, and inform future research and policy development aimed at reducing attrition rates in Bachelor of Nursing programs. This scoping review will be guided by the following research questions:What are the most common reported factors contributing to attrition among students enrolled in Bachelor of Nursing programs?What strategies have been implemented or proposed to improve retention in bachelor-level nursing education?What gaps exist in the current literature regarding nursing student attrition and retention strategies at the bachelor level?

To address these questions, this scoping review categorizes the factors contributing to attrition and retention, providing a structured approach to understanding nursing student persistence. By synthesizing academic and non-academic strategies, it offers a comprehensive perspective on evidence-based retention approaches. The review enhances the understanding of nursing student attrition and retention through these contributions, providing valuable insights for educators, institutions, and policymakers to strengthen nursing education outcomes.

## 2. Materials and Methods

This study was conducted as a scoping review, which is a suitable method for mapping the available evidence and analyzing knowledge gaps related to the topic of interest. The review followed the Preferred Reporting Items for Systematic Reviews and Meta Analyses (PRISMA 2020) Checklist [13] (Appendix A) and the Joanna Briggs Institute (JBI) methodological framework [14] to systematically map the literature on nursing student attrition and retention strategies. The primary aim was to identify contributing factors, synthesize existing evidence, highlight knowledge gaps, and establish a foundation for future research and intervention strategies. The methodology adhered to the five-stage framework by Arksey and O’Malley, further developed by Levac et al., which includes the following stages: identifying the research question; identifying relevant studies; selecting the studies; charting the data; and collating, summarizing, and reporting the results [15,16]. The review was not registered in a systematic review database.

### 2.1. Identifying the Research Question

The Joanna Briggs Institute recommends the use of the Population/Concept/Context (PCC) framework to guide the development of review questions in scoping reviews. In this study, the population is undergraduate or bachelor-level nursing students enrolled in formal nursing education programs; the concept includes attrition/dropout; and the context is academic or clinical education environments relevant to bachelor nursing programs across diverse countries. The term attrition in nursing education is commonly used to describe students who leave or discontinue their studies before completing the program and is often used interchangeably with related terms such as dropout, withdrawal, or non-completion. However, in this study, we preferred to use attrition throughout the article, except when referring to studies that specifically used the term “dropout.” Based on this framework, the following review question was developed: “What is the scope of the literature on factors contributing to attrition and strategies for improving retention in Bachelor of Nursing programs?”

### 2.2. Identifying Relevant Studies

#### Data Sources and Search Strategy

A comprehensive systematic search was conducted across six databases: MEDLINE/PubMed, Embase via OVID, PsycInfo, and the Cumulative Index to Nursing and Allied Health Literature (CINAHL). Relevant subject headings (e.g., MeSH) and free-text terms were used, combined with Boolean operators (AND, OR) to refine the search. Truncation and wildcard symbols were applied where necessary to capture variations in terminology (Table 1). A university librarian assisted in developing the search strategy.

### 2.3. Study Selection

Inclusion and exclusion criteria were carefully defined to obtain specific and relevant evidence.

#### 2.3.1. Inclusion Criteria

The following inclusion criteria were applied:Studies focused on students enrolled in bachelor-level (BSN or equivalent) nursing programs of 3–4 years’ duration.Studies addressing attrition-related outcomes such as dropout, intention to leave, or retention.Peer-reviewed empirical studies (qualitative, quantitative, or mixed-methods).Published in English between 2010 and 31 December 2024.Conducted in any country to capture global patterns.

#### 2.3.2. Exclusion Criteria

The aim of this review was to explore attrition specifically within bachelor-level nursing programs. Therefore, the following exclusion criteria were applied:Studies focusing on non-nursing students, or nursing education at diploma, associate, or graduate levels.Grey literature, unpublished studies, editorials, commentaries, protocols, letters, or abstracts without primary data.Studies not addressing attrition, dropout, or retention.Articles not available in English.

The database search was initially conducted in May 2023 and subsequently updated in February 2025 to ensure that the review reflects the most current and relevant studies published up to the end of 2024. Relevant studies were selected, and duplicate records were identified by exporting all studies to the EndNote reference manager (Version 21). A total of 1416 records were identified through database searches. After removing duplicates, 1238 records remained for title and abstract screening. Of these, 168 reports were selected for full-text review, and 162 were excluded due to irrelevant population, irrelevant outcomes, or non-eligible language. Following the full-text screening, 6 new studies were added. In total, 19 studies met the inclusion criteria and were included in the final review. Figure 1 presents the updated PRISMA flowchart for this review.

### 2.4. Data Charting and Extraction

A two-stage data-screening process was implemented to ensure accuracy and reliability. Title and abstract screening were conducted independently by two reviewers (RLS and DMP), with discrepancies resolved through discussion. Full-text review and data extraction were conducted by two independent teams (RLS/DMP and ARG/SK), with disagreements resolved by a third reviewer (LIMH). Data extraction focused on authors, publication year, journal, study aim, research question, population, methods, analysis, and key findings related to attrition and retention. Studies that lacked a precise definition of attrition were reviewed for contextual relevance before inclusion. We pilot-tested the data extraction form using two articles, independently reviewed by the researchers, to ensure consistency, clarity, and relevance in capturing key information aligned with the review objectives. Pilot testing helped resolve ambiguities and refining the extraction form, ensuring a shared understanding among reviewers and enhancing the reliability and transparency of data charting.

### 2.5. Collating, Summarizing, and Reporting the Results

A structured content analysis approach [17] was employed to synthesize findings from both qualitative and quantitative studies. Using NVivo 14, extracted data were categorized into thematic clusters based on the research questions. The authors reached consensus on the thematic categorization of factors associated with attrition and the strategies suggested for retention. Attrition factors (RQ1) were categorized into four key themes: academic factors, institutional and social support, personal factors, and economic challenges. Suggested retention strategies (RQ2) were classified into academic and non-academic approaches, highlighting interventions shown to be effective in reducing nursing student attrition.

### 2.6. Quality Appraisal

Scoping reviews aim to map the breadth and characteristics of available evidence rather than to evaluate the methodological quality of individual studies [18]. Therefore, we did not conduct a formal quality assessment of the included papers.

## 3. Results

### 3.1. Characteristics of the Studies

The electronic database search retrieved 1416 studies. After full-text screening, 19 studies were included in the review. Most articles were excluded because they were not relevant to the research topic or focused on the wrong population. The studies employed various designs, with quantitative studies being the most common. Quantitative methods were used in retrospective analyses [19,20], prospective cohorts [21,22], and cross-sectional studies [23,24,25,26]. Data collection methods included semi-structured interviews [3,27,28], face-to-face or virtual interviews [29,30], telephone interviews [31,32], and questionnaires and scales [21,22,23,25,26,33]. The sample sizes ranged from 10 to 759 students, and mean ages generally ranged from 19 to 31 years. Most studies were published between 2020 and 2024, reflecting increased research interest in nursing student attrition in recent years. Studies were conducted across diverse locations, including the USA (3), the Netherlands (3), Italy (3), Iran (2), Belgium (2), South Africa (1), Denmark (1), Finland (1), Canada (1), Spain (1), and the UK (1), mostly within university-based or college-level nursing programs.

The main outcomes investigated included academic performance, self-esteem [33], psychological distress, psychosocial stressors [21,23], perceived institutional and social support [22], and personal motivations or factors related to dissatisfaction [28,32,34] (Table 2).

### 3.2. Factors Associated with Attrition

This review identified multiple interrelated factors contributing to attrition among bachelor nursing students, categorized into the following thematic groups: academic factors, institutional and social support, personal factors, and economic challenges (Table 3).

#### 3.2.1. Academic Factors

Academic factors were reported in 14 studies. Common issues included poor academic preparation in high school, low scores on national entrance exams (pre-entry exams), challenges with meeting program entry requirements, insufficient study skills, and lower academic performance or GPA [19,29,30,33,37]. Excessive academic workload, unclear expectations of nursing programs, difficulties in integrating theoretical knowledge into clinical settings, and lack of language proficiency were also found to influence attrition [3,28,29,34,35,36].

#### 3.2.2. Institutional and Social Support

Institutional and social factors were the most frequently reported category (16 studies), commonly described as limited faculty support [27,30], inadequate facilities or poorly equipped nursing skills laboratories, geographic limitations, and negative social climates within institutions [3,27,28,32]. Additional issues included weak clinical supervision [32], social exclusion or a low sense of belonging [25], and rigid program structures [34,35].

#### 3.2.3. Personal Factors

Personal factors contributing to attrition included low self-esteem, emotional distress, lack of professional identity, and difficulty managing personal responsibilities [23,26,30,33,35]. Other significant predictors were male gender, older age at entry, physical and psychological health issues, and a mismatch between personal expectations and professional realities [20,22,29,31,34]. Family circumstances, mental health concerns, and prior academic performance also played a role in decisions to withdraw [20,27].

#### 3.2.4. Economic Challenges

Economic and financial stressors were less frequently reported but remained relevant. Students experiencing financial difficulties, such as high tuition costs [27,30,31,35], the need to work long hours while studying, or a lack of scholarships or financial aid, were at higher risk of leaving the program [27].

### 3.3. Strategies for Retention

Retention strategies identified in the review were classified into academic and nonacademic strategies (Table 4).

#### 3.3.1. Academic Strategies

Several studies highlighted the importance of early academic support, skill-building, and providing additional academic assistance through tutoring, mentoring, and faculty counseling [3,19,30,31,33,35,36]. Developing realistic and supportive curricula and intervening early when academic challenges arise are central to improving retention [28,34,36]. Additionally, peer-to-peer mentoring, social integration, and courses focused on coping skills, study techniques, time management, and critical thinking were found to support student retention [3,26,35,36].

#### 3.3.2. Nonacademic Strategies

Non-academic strategies were diverse and frequently emphasized. Key interventions included improving faculty-student relationships and fostering a culture of empathy [27,30,32], enhancing resilience and coping abilities [23,26], and promoting peer mentoring and learning communities to strengthen students’ sense of belonging [25,36]. Early identification of at-risk students based on psychosocial factors enabled the provision of tailored support [23,33], while involving families was proposed to reinforce student persistence [36]. Financial stress emerged as a critical barrier, highlighting the need for flexible scheduling and financial support mechanisms [31,34].

### 3.4. Research Gaps, Future Directions, and Practical Implications

Table 5 summarizes critical gaps, future research directions, and implications for practice. The included studies revealed several recurring gaps in the literature, including limited understanding of psychological resilience, self-esteem, and emotional challenges among nursing students [19,32,33]. A lack of standardized definitions for attrition hinders comparability across studies [27,30]. Many studies were limited to single institutions or narrow settings, which affects the generalizability of their findings [22,32,36]. There is also an underrepresentation of at-risk student populations, including those with family responsibilities, financial challenges, or lower entry qualifications [20,31].

Studies suggested that future research should focus on conducting longitudinal studies to explore resilience, self-esteem, psychological distress, and academic stressors [19,21,23]; evaluating the effectiveness of mentorship and emotional support systems [3,30]; exploring early academic experiences, student motivations, and perceptions of the nursing profession [30,35]; and examining strategies to enhance resilience and emotional well-being among nursing students facing emotional distress [26,32].

Practical implications include developing emotional and resilience support programs [19,23], improving mentorship and faculty training, enhancing social integration and belonging, addressing bullying, providing targeted financial and academic assistance, and implementing early interventions tailored to at-risk students [25,28,30,32,33,34,35].

**Table 5 nursrep-15-00182-t005:** Summary of research gaps, future directions, and practical implications.

Author(s), Year, Country	Identified Gaps	Suggestions for Future Research	Implications for Practice
Abele et al., 2013, USA [19]	Lack of studies examining non-nursing courses (e.g., psychology) as predictors of success among at-risk nursing students Limited research focusing specifically on academically probationary students in nursing programs	Explore the role of critical thinking development via interdisciplinary course collaboration in improving nursing student outcomes	Monitoring course performance (e.g., psychology) and identify at-risk students early for intervention Implementing mentorship, student-to-student support, and critical thinking courses to improve retention and academic success
Ashghali Farahani et al., 2017, Iran [29]	Lack of preparation and awareness before entering nursing education Discrepancy between expectations and realities in both theoretical and clinical education Lack of support and professional identity Clinical settings not prepared to support student learning Poor student supervision and workforce planning	Explore institutional interventions and policy changes to reduce attrition Longitudinal research is needed to examine the long-term impact of clinical experiences on student retention	Enhance pre-nursing career guidance Improve theoretical and clinical coordination Strengthen faculty training and supervision Promote a supportive and respectful learning environment in clinical practice Address gender-specific challenges and professional identity formation
Bakker et al., 2021, The Netherlands [21]	Limited longitudinal studies examining the effects of psychosocial work characteristics on nursing student attrition Lack of research on changes in distress and intention to leave over time Lack of research on the impact of offensive behaviors such as workplace violence on nursing student distress and dropout Need for further exploration of protective factors such as co-worker and supervisor support in clinical settings	Longitudinal studies to assess the long-term impact of workplace violence and psychological demands on student dropout Development of interventions aimed at improving the psychosocial work climate in clinical placements Exploring the role of faculty and organizational policies in mitigating distress and dropout among nursing students	Improve the psychosocial work environment of nursing students. Enhancing co-worker and supervisor support to reduce nursing students’ intention to leave Reducing workplace violence and psychological demands in clinical placements Improve co-worker support alongside supervisor support Attention should be given to nursing students’ psychological strain and exposure to violence during clinical placements
Barbé et al., 2018, USA [35]	Limited data on early predictors of attrition at the end of the first semester There is a need for improved identification of at-risk students from diverse backgrounds	Examining how self-perceptions of nursing students impact attrition and what strategies support confidence and persistence Examining whether overlapping factors can be combined into a risk index to improve prediction and guide targeted interventions	Systematic attention should be given to social determinants among students in nursing programs Early identification of at-risk students using academic and psychosocial indicators Development of support programs targeting English language support, financial aid, and confidence-building measures for minority students
Canzan et al., 2022, Italy [3]	Limited data on the effectiveness of mentorship programs Lack of studies on students who considered leaving but stayed	Further exploration in other nursing academic settings is needed in order to give a deep understanding of the nursing student attrition Exploring the effectiveness of strategies to improve nursing students’ intention to stay	Strengthen mentorship initiatives in nursing education programs
Dancot et al., 2021, Belgium [33]	Self-esteem is rarely measured at the start of nursing education The link between self-esteem, state anxiety, self-efficacy and dropout has been underexplored Further research of self-esteem and dropout using Mruk’s two-dimensional self-esteem is suggested	Conduct longitudinal, mixed-methods studies to explore self-esteem dynamics over time Model the system of factors influencing self-esteem and dropout Compare nursing students with other student populations. Further explore the relevance of self-esteem profiles	Institutions should support student self-esteem early on, especially for those with anxiety or low self-efficacy Improve communication and support systems to foster a sense of belonging Follow first-year nursing students monthly Consider self-esteem in dropout prevention efforts
Kox et al., 2022, The Netherlands [22]	Lack of qualitative insights on student dropout Unclear causal link between intention to leave and actual dropout	Explore interventions that foster a supportive workplace culture Conduct qualitative studies to explore reasons for dropout and intention to leave Examine gender-related dropout risks, especially among male students Investigate the impact of severity of musculoskeletal complaints Systematic exit interviews or surveys with students that have decided to quit nursing education	More attention should be paid to the students’ personal circumstances during nursing education Provide early support for students at risk (e.g., males, those with high distress) Promote co-worker support and decision-making autonomy in clinical placements Offer physical workload and ergonomic training early in nursing education
Kukkonen et al., 2016, Finland [27]	Little knowledge on the long-term effects of early intervention programs Lack of a common definition and tracking method for attrition Insufficient identification and support for at-risk students	Not reported	Introduce early interventions that prevent student attrition Schools should create models to recognize and support at-risk students through tailored interventions
Matteau et al., 2023, Canada [23]	Limited understanding of how academic conditions influence psychological distress and intention to leave Lack of longitudinal studies on academic stressors and student attrition Lack of studies examining effort-reward imbalance or school-work–life conflict among nursing students	Conduct longitudinal studies to establish causal relationships between academic stressors and attrition Develop and test interventions to reduce school-work–life conflicts and modulate workload. Engage nursing students and faculty in participatory research to identify context-specific challenges Explore overcommitment because of academic workload in nursing education	Implement interventions targeting modifiable academic conditions (e.g., reduce workload, improve work–life balance, increase perceived rewards) improving nursing students’ mental health and retention
Mazzotta et al., 2024, Italy [30]	lack of insight into attrition across different institutional and cultural contexts	Conduct research with larger samples in varied educational and cultural contexts to validate and extend findings	Provision of adequate support systems, mentorship, and resources for students Enhance the quality and relevance of clinical learning experiences Introduce financial assistance programs for economically disadvantaged students
Roos et al., 2016, South Africa [31]	Limited research on nurses’ career satisfaction over time	Analyze long-term career satisfaction and its impact on retention Conduct more detailed and multi-site investigations into the reasons for nursing student attrition in South Africa	Provide career development programs to sustain job satisfaction Strengthen academic and financial support systems; implement wellness interventions and structured orientation programs to improve retention
Roso-Bas et al., 2016, Spain [24]	Limited studies focusing on emotional predictors of dropout in nursing students Lack of research on protective emotional factors	Longitudinal studies focusing on the evaluation of emotional variables like optimism and emotional regulation affecting dropout	Integrate emotional intelligence training and psychological support into nursing curricula to reduce dropout risk
Sharif-Nia et al., 2023, Iran [25]	Limited research on the impact of bullying behaviors on nursing students’ sense of belonging and academic satisfaction Faculty and clinical instructors’ contribution to bullying in nursing education. Interventions that effectively mitigate bullying and promote student retention	Examine longitudinal impacts of bullying on attrition Effectiveness of intervention programs that enhance student belongingness and major satisfaction to reduce dropout rates	Implement anti-bullying policies that target faculty behavior and clinical instructor interactions Enhance nursing students’ sense of belonging through mentorship programs and peer support networks
Soerensen et al., 2023, Denmark [32]	Inadequate preparation for the emotional challenges of clinical placements Social exclusion and lack of belonging were underexplored as dropout factors	Further studies should explore strategies to enhance emotional support and resilience among nursing students Investigate the development of student resilience and the educator’s role in strengthening it. Compare students who dropped out with those who stayed despite similar experiences.	Improve clinical guidance and social inclusion Implement interventions to support students facing emotional and personal stress Foster caring, supportive relationships between educators and students to develop professional identity Create emotionally safe clinical and academic environments that support reflection and resilience
Ten Hoeve et al., 2017, The Netherlands [28]	Lack of robust data on why Dutch nursing students consider leaving pre-registration nursing programs. Limited insight into how training organization, quality, and staff support affect dropout rates Insufficient understanding of the impact of team support and integration in clinical placements on student retention	Further qualitative research to better understand student experiences with training programs and clinical placements Examine strategies to reduce theory-practice gap and improve academic-practical integration Investigate the role of team dynamics and student integration into clinical teams	Strengthen cooperation between teaching staff and clinical mentors to support students effectively Improve the structure and content of training programs, ensuring consistency in quality and expectations Recognize and nurture intrinsic motivations while addressing external barriers like poor mentorship or unclear career expectations
Van Hoek et al., 2019, Belgium [26]	Insufficient analysis of resilience impact on academic success and attrition	Investigate the predictive value of resilience on long-term success Investigate causal pathways between resilience, mental health history, and dropout Evaluate targeted interventions	Enhance resilience training to support student academic achievement
Viottini et al., 2024, Italy [34]	Limited research on the link between motivations for enrolment and dropout among first-year nursing students Few studies combining quantitative and qualitative methods to understand dropout factors Limited studies focus on first-year students or use longitudinal designs	Conduct longitudinal, multicenter studies to analyze dropout trends across different universities Explore effectiveness of interventions aimed at students who enroll in nursing as a second choice Explore strategies to enhance professional identity and belonging among first-year nursing students.	Implement targeted interventions for students who enroll in nursing as a second choice Introduce interventions like peer support, time management training, and mental health strategies Enhance clinical placement experiences to align expectations with real-world nursing practice
Williams, 2010, USA [36]	Lack of understanding about how personal mindset and connection-building influence persistence in nursing programs	Examine interventions that enhance early nursing student persistence Conduct multi-site studies on how student engagement with persistence-focused interventions affects retention and graduation	Develop faculty-driven strategies to improve student persistence Create structured opportunities to build student-to-student and student-faculty connections, engage families, support mindset development, and target key stress points early in the program
Wray et al., 2017, UK [20]	Inadequate data on factors influencing nurse program completion Limited understanding of how demographic factors like age, dependents, and residency status impact attrition risk	Analyze institutional and personal factors affecting completion rates Explore how individual student characteristics interact with institutional support to influence progression	Establish institutional policies that support student success Early identification of students at risk (e.g., younger, non-local, no dependents) Tailor support to diverse student needs

## 4. Discussion

This scoping review aimed to systematically map the available evidence and explore research gaps related to the factors contributing to attrition and the proposed retention strategies in bachelor nursing education. The findings indicate that attrition among nursing students is a complex issue, requiring student-centered retention strategies to reduce attrition and support student success across diverse educational contexts. Consistent with prior research, nursing student attrition rates are influenced by both individual circumstances and institutional factors within educational institutions [3,37,38].

### 4.1. Academic Challenges

Academic preparedness is a significant predictor of nursing student success. Consistent with recent findings, our review identified that inadequate academic preparation, including insufficient study skills and lower academic performance, significantly contributes to nursing student attrition. Research has consistently shown that pre-nursing science GPA and overall college GPA are critical indicators of program completion [39,40]. High academic demands, coupled with difficulties in integrating theoretical knowledge into clinical practice, contribute to student stress and disengagement, ultimately leading to higher attrition rates [12]. Students struggling with low entry qualifications, weak study habits, and time management issues are particularly at risk of attrition [41]. Additionally, poor alignment between student expectations and the realities of clinical practice has been cited as a reason for early withdrawals [42]. Nursing curricula are often rigorous and time-intensive, combining theoretical coursework with clinical placements [43]. This demanding workload leaves students with limited time and energy to cultivate social networks and engage in peer or institutional support activities [3,33]. Studies indicate that insufficient social support is one of the most critical non-academic predictors of attrition [25,30]. When academic responsibilities dominate students’ time, they may experience social isolation, reduced sense of belonging, and increased psychological distress—factors known to undermine motivation and resilience [23,26]. Ten Hoeve et al. (2017) found that nursing students who lacked supportive social environments due to academic overcommitment were more likely to report withdrawal intentions [28]. This gap between theoretical learning and practical application highlights the need for stronger academic support systems, enhanced faculty–student engagement, and improved clinical preparedness programs.

### 4.2. Institutional and Social Support

The role of institutional support in student retention cannot be overstated. Studies have shown that students with access to mentorship, tutoring, and a strong sense of belonging are more likely to persist in their programs [44,45]. Insufficient clinical placement sites, faculty shortages, and limited access to essential resources create additional barriers to student success. Nursing education relies heavily on clinical training, and negative experiences during placements, such as high stress levels, poor supervision, and lack of hands-on learning opportunities, can significantly impact retention [46,47]. A UK study reported that 40% of nursing students who considered dropping out cited negative clinical placement experiences as a key factor [12]. Other studies confirm that clinical learning challenges, including conflicts with peers and healthcare staff, fear of harming patients, and lack of clinical skills, contribute to student dissatisfaction [43,48]. Our review extends the existing literature by emphasizing the effectiveness of comprehensive retention strategies that integrate academic and non-academic support, including prioritizing structured mentorship programs, faculty–student engagement, and improved supervision during clinical training. A recent scoping review by Everett (2020) supports this approach, highlighting that successful retention strategies attend to social and academic integration [49]. However, faculty shortages remain a significant barrier to implementing these strategies in nursing education. Limited educator availability reduces individualized support, mentorship, and timely feedback—factors linked to persistence [30,50]. Overburdened faculty often lack the capacity to address academic or emotional challenges, which may lead students to feel unsupported and consider dropping out [27,36]. In clinical settings, faculty shortages hinder adequate supervision, making it harder for students to apply theoretical knowledge and build confidence [3]. This is particularly challenging for underrepresented students [25]. Therefore, the WHO (2020) emphasizes the need to invest in faculty recruitment and retention to address the global nursing workforce shortage and reduce attrition in nursing programs [51].

### 4.3. Professional Identity and Perceptions of Nursing as a Career

A student’s decision to pursue nursing and their perceptions of professional suitability are key determinants of attrition. Tinto’s (1975) integration model suggests that academic and social integration play critical roles in degree completion [52]. Students with low entry qualifications or a weak professional identity often find it harder to integrate into clinical settings, increasing their risk of attrition [53]. Several studies highlight the importance of aligning nursing curricula with real-world clinical experiences to enhance professional identity and career readiness [54,55]. However, more research is needed to explore how different teaching models influence the formation of professional identity and how early career exposure affects long-term retention.

### 4.4. Academic Support, Mentoring, and Student Persistence

In accordance with other studies on this topic, this review found that students’ motivation, resilience, and self-confidence are key personal factors influencing program completion [7]. Resilience refers to a person’s ability to cope with stress and difficult situations, allowing them to maintain mental strength [56]. Defined as the ability to adapt well to adversity, resilience helps students manage stress and overcome challenges. Nursing students often face high stress due to long clinical hours, exposure to patient suffering, and academic demands [57]. Recent studies link resilience to academic success, which is influenced by factors such as health, family support, motivation, and financial resources [57,58]. Educators should implement resilience-promoting interventions to better support student success [57].

This review identified several key educational factors influencing retention, including challenges with group work, uncertainty about academic performance, language barriers, and deficiencies in study skills. Prior research confirms that poor study habits and ineffective time management negatively impact retention [41]. However, structured academic support services, such as tutoring and faculty mentoring, significantly improve retention rates [59]. Peer mentoring has also been found to enhance critical thinking, learning experiences, and student well-being [60,61,62]. A large-scale study involving 4472 undergraduate nursing students found that those who sought academic support were over seven times more likely to persist in their programs [63]. These findings highlight the need for early identification of at-risk students and the expansion of tutoring and mentoring services to improve retention.

Support strategies must be tailored to diverse student backgrounds. Psychological services, support groups, and online resources are particularly beneficial for students from various cultural and linguistic contexts [63,64]. Family and peer support, providing emotional, financial, and practical assistance, plays a vital role in encouraging students to persist in demanding courses [65,66]. These strategies also help promote confidence and motivation, thereby fostering retention [63].

Nursing education requires considerable commitment and hard work. Employing effective study strategies and techniques, which are not included in most nursing curricula, can be key to achieving success. Incorporating time management, problem-solving, critical thinking, and communication skills into nursing curricula can enhance retention [65,67]. Time management and problem-solving skills, in particular, were found to be associated with higher academic success [68].

### 4.5. Personal Responsibilities, Gender Differences, and Attrition Risks

Balancing family obligations, work schedules, and academic demands significantly affects students’ ability to complete their programs. Gender disparities in attrition rates have been observed in multiple studies, with male nursing students showing a significantly higher attrition rate than females [53,69,70]. However, conflicting evidence exists regarding the role of age at program entry in predicting attrition risk. Some studies suggest that older students are more likely to persist due to better-developed skills, elevated confidence, and more informed career choices [71], while others indicate that younger students may have better cognitive abilities, fewer external responsibilities, and less involvement in social relationships—particularly romantic ones—reducing their likelihood of leaving [72]. Future research should focus on understanding how demographic variables interact with institutional and academic factors to influence retention.

### 4.6. Clinical Support and Learning Environments

Adequate clinical support, access to well-equipped skills laboratories, and a supportive learning environment are essential for preventing attrition. Students lacking clinical preparedness often experience fear, anxiety, and low self-confidence, which increases their risk of withdrawal [43]. One-on-one mentorship and preceptor guidance have been found to enhance learning experiences, boost student confidence, and improve retention [43,73]. However, faculty shortages and resource limitations hinder the availability of high-quality clinical training [74]. While high-fidelity simulation models have been shown to bridge clinical learning gaps [75], many institutions struggle to fund and maintain advanced simulation technology. Future studies should explore the long-term impact of simulation-based education on clinical competency and retention rates.

### 4.7. Research Gaps and Implications

This review reveals a clear need to standardize definitions of attrition to improve cross-study comparisons. Many studies were conducted within single institutions or lacked longitudinal follow-up, limiting the generalizability of their findings. Additionally, the psychosocial dimensions of attrition, such as bullying, emotional fatigue, and lack of professional identity, remain underexamined. Future research should prioritize longitudinal and mixed-methods studies that capture the trajectory of student experiences, particularly among underrepresented groups [20,32]. Intervention-based studies are urgently needed to evaluate the effectiveness of targeted strategies aimed at improving retention and reducing attrition among nursing students.

To guide future research on nursing student attrition, we suggest adopting the “wicked problem” framework proposed by Hamshire et al. (2019) [76]. Attrition is not a single-issue challenge [77] but the outcome of multiple interacting factors across personal, academic, clinical, and institutional systems. This approach encourages researchers to move beyond linear models and consider the complex, context-dependent nature of student experiences. By applying a systems-based, stakeholder-informed conceptual model, future studies can better explore how these interrelated influences shape attrition and can develop more holistic and adaptable retention strategies.

### 4.8. Practical and Policy Recommendations

Nursing programs should adopt a proactive, multifaceted approach to student retention. Academic interventions must be complemented by structures that promote emotional well-being, such as mentorship programs, mental health resources, and social inclusion initiatives. Faculty training should emphasize supportive pedagogy and the early identification of students in distress. Additionally, financial support and flexible academic pathways can help reduce attrition risks, particularly for mature and working students. Creating a “culture of retention” within institutions—where challenges are normalized, and support is readily accessible—is critical for retaining future nurses. Interventions should not only respond to student struggles but also anticipate them, especially during clinical transitions or after academic setbacks.

While most of the studies originated from high-income countries, particularly in Europe, this geographic concentration may limit the applicability of our findings to diverse cultural and educational contexts. Educational structures, student support systems, and healthcare workforce expectations vary considerably across regions, particularly in low- and middle-income countries, where resource constraints and differing sociocultural expectations may influence nursing student attrition in unique ways [51,77]. For instance, barriers such as financial hardship, limited institutional support, and social norms regarding gender and caregiving roles may compound educational challenges. While social support was a dominant theme across studies, its delivery and effectiveness may vary significantly in contexts where formal support systems are less established. Similarly, clinical placement challenges, educational resources, mentorship practices, and faculty–student ratios may differ considerably in low-resource settings [51]. Therefore, while the identified themes offer valuable insights into attrition factors, future research should explore how these factors manifest in underrepresented contexts to better inform globally relevant retention strategies.

### 4.9. Strengths and Limitations

This review has several limitations. First, the small number of included studies and their focus on specific countries limit generalizability. Second, the exclusion of grey literature, unpublished studies, and studies published before 2010 may have omitted valuable insights. Third, no formal quality assessment was conducted, in line with JBI guidelines for scoping reviews, as such reviews do not typically require one. Although a formal appraisal was not undertaken, we observed considerable diversity in methodological approaches, including variations in study design and sample size. Many studies employed qualitative or exploratory methods; while some included large samples, others relied on smaller sample sizes. This variability should be considered when interpreting the patterns and trends identified in this review, as it may influence the scope and depth of the available evidence. Fourth, most of the included studies were conducted in high-income countries, which may reduce the transferability of findings to culturally diverse or resource-constrained educational contexts. Fifth, the wide variation in reported attrition rates reflects inconsistencies in how studies define attrition—some considered only voluntary withdrawals, while others included academic failure or institutional dismissals. These inconsistencies underscore the need for standardized definitions in future research. Finally, language restrictions may have excluded relevant studies published in non-English languages.

Despite these limitations, this review significantly contributes to understanding nursing student attrition. It categorizes the factors contributing to attrition into four themes, providing a structured approach for identifying risk factors. The findings emphasize the importance of accurate attrition rate reporting and the need for targeted retention strategies. Strengths of this review include its comprehensive, up-to-date synthesis of the literature and its multinational scope.

## 5. Conclusions

This scoping review confirms that attrition in bachelor-level nursing education is a persistent challenge, driven by a combination of academic demands, institutional shortcomings, personal stressors, and financial constraints. Although numerous studies have explored the factors influencing attrition, inconsistencies in definitions and methodologies hinder cross-comparability and the development of a cohesive evidence base. Common and modifiable contributors include the learning climate, faculty engagement, students’ sense of belonging, and economic hardship. While awareness of these issues is increasing, coordinated, evidence-informed strategies to enhance retention remain limited and inconsistently applied. Academic support systems, inclusive learning environments, and faculty development programs that promote empathy and student-centered engagement show promise. Equally important are non-academic interventions, such as psychological support, mentoring, and financial assistance, tailored to the diverse needs of today’s nursing students. However, these approaches are often under-researched and not widely implemented.

Future research should prioritize intervention-based studies that evaluate scalable, context-sensitive solutions, particularly for underrepresented and at-risk student populations. Ensuring equity and effectiveness in retention strategies requires a deeper understanding of how students navigate nursing education. Moreover, future research and retention strategies should also consider how both perceived and actual working conditions—during clinical placements and in professional practice—may contribute to nursing student attrition and broader workforce challenges. To build a resilient nursing workforce and safeguard the quality of healthcare, nursing education must evolve—not only through curricular reform but also through the creation of supportive, proactive environments that promote student success, reduce attrition, and enhance well-being from enrollment through to graduation.

## Figures and Tables

**Figure 1 nursrep-15-00182-f001:**
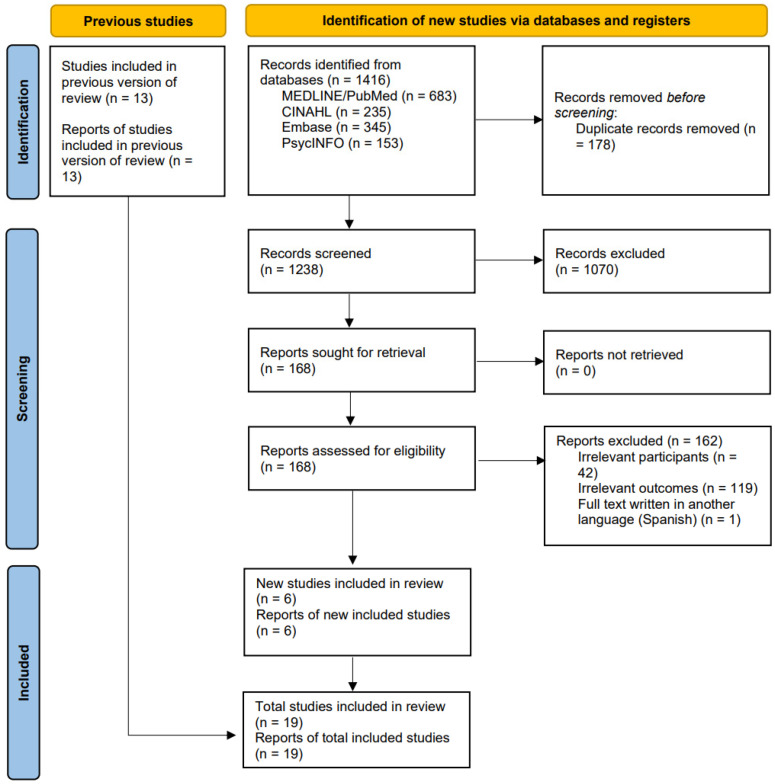
PRISMA Flowchart.

**Table 1 nursrep-15-00182-t001:** Search terms.

Type of Search Terms	Search Terms	Boolean Operators
Population	Students, Nursing/Students	
MeSH/CINAHL headings	undergraduate/Students, Nursing, Practical/Baccalaureate Nurses/Education, Nursing, Baccalaureate (MM “Students, Nursing”) OR (MM “Students, undergraduate Nursing”) OR (MM “Students, Nursing, Baccalaureate”) OR (MM “Students, Nursing, Practical”) OR (MM “Baccalaureate Nurses”) OR (MM “Education, Nursing, undergraduate”)	^1^ OR ^2^ OR AND
Freetext keywords	Student nurs*, nurs* students, undergraduate nurs*, baccalaureate nurs*	
Intervention MeSH/CINAHL headings	Education, Baccalaureate/Education, Clinical/Teaching Methods, Clinical/Teaching Materials, Clinical/Practical Nursing (MM “Education, Baccalaureate”) OR (MM “Education, Clinical”) OR (MM “Teaching Methods, Clinical”) OR (MM “Teaching Materials, Clinical”) OR (MM “Practical Nursing”)	^1^ OR ^2^ OR AND
Freetext keywords	Education*, program design, curricul*, teaching method, training, practic*, teach*	
Outcome MeSH/CINAHL headings	Students attrition/Academic failure/Academic Performance/academic achievement (MM “Student attrition”) OR (MM “Academic Failure”) OR (MM “Academic Performance”)	^1^ OR ^2^ OR AND
Freetext keywords	Attrition, dropout, retention, academic failure, leave, discontinuation, withdrawal	

**Table 2 nursrep-15-00182-t002:** Characteristics of the included studies (n = 19).

Author(s), Year, Country	Study Design	Method	Aim	Sample (n) and Mean Age	Setting	Concept	Outcome
Abele et al., 2013, USA [19]	Exploratory retrospective study	Students’ records	Identify courses that may predict student success in completing the baccalaureate nursing program (BNP) and examine factors associated with attrition among nursing students on academic probation.	302 students—27.2 years	University nursing school	Attrition	Course failures Demographic characteristics (Age, gender, ethnicity, program type)
Ashghali Farahani et al., 2017, Iran [29]	Descriptive, qualitative	Face-to-face interviews, focus group interviews Participant observation	Elucidate the factors that lead students to drop out or express a willingness to drop out, as perceived by the students.	19 students (intended to leave or already left the program)—23.4 years	Three BNP	Dropout Intention to leave	Attrition factors—before and after admission to the program
Bakker et al., 2021, The Netherlands [21]	Prospective cohort study	Baseline, second semester and one-year follow-up questionnaire, the Distress Screener, data from student administration about the dropout status	Investigate associations between psychosocial work characteristics and distress and intention to leave nursing education among third-year nursing students.	363 third-year nursing students—24 years	BNP at a University of Applied Sciences	Intention to leave Actual dropout	Supervisor support Co-worker support Psychosocial distress Exposure to violence
Barbé et al., 2018, USA [35]	Descriptive, comparative	Administrative database Web-based survey The Educational Requirements Subscale	Identify demographic, academic, and social determinants associated with attrition at the end of the first semester	164 students—24 years	Upper division BNP	Attrition	Demographic factors (Age, Gender Race/Ethnicity) Academic factors Social factors (Economic stability, Education, Social and community context, Health and health care, Neighborhood and built Environment)
Canzan et al., 2022, Italy [3]	Descriptive, qualitative	Semi-structured interview	Investigate the reasons behind nursing students’ attrition	31 students—21 years	Three-year BNP	Dropout	The reasons behind drop-out Physiological causes
Dancot et al., 2021, Belgium [33]	Cohort study	Questionnaire at start of program and academic records at 1-year follow-up	Describe first-year nursing students’ self-esteem prior to the influence of nursing education and explore its relationship with dropout.	464 students; Median age: 19 years	BNP	Dropout	Self-esteem Dropout
Kox et al., 2022, The Netherlands [22]	Prospective cohort	10-point Likert scale Registered data	Explore the determinants of intention to leave nursing education and actual dropout from nursing education	711 third-year students—23.5 years	Three-year BNP	Physical work factors Actual Dropout Intention to leave	Sociodemographic characteristics Physical work factors Musculoskeletal complaints at baseline Psychosocial factors
Kukkonen et al., 2016, Finland [27]	Descriptive, qualitative	Semi-structured interview	Describe the discontinued student in nursing education and the student’s own experiences of reasons for leaving nursing school	25 students—31 years	Two different universities of applied sciences	Dropout	Characteristics of discontinued students The student’s own experiences of reasons for leaving nursing school
Matteau et al., 2023, Canada [23]	Cross-sectional correlational study	Self-administered online questionnaire, Poisson robust multivariate regression models	Explore the associations between academic conditions and (1) psychological distress and (2) intention to leave school among nursing students.	230 nursing students (131 from Cegep, 99 from university), Cegep—22.7 years, University 29.3 years	Two nursing schools: Cegep (publicly funded college) and university (bachelor’s degree)	Intention to leave	Psychosocial stressors, Intention to leave
Mazzotta et al., 2024, Italy [30]	Qualitative descriptive study using thematic analysis	Face-to-face virtual interviews	Explore perceptions of nursing students and directors of Bachelor of Nursing degree courses regarding reasons for attrition among nursing students.	12 Students—24.8 years	Bachelor of Nursing programs at one Italian university	Attrition was defined as the number of students enrolled in a nursing program who did not complete it	Identified reasons for attrition
Roos et al., 2016, South Africa [31]	Descriptive, quantitative	Structured telephonic interview	Determine attrition rate and factors influencing undergraduate students to discontinue their nursing studies	54 students—21.3 years	Three South African Universities	Attrition	Attrition rates Factors leading to attrition
Roso-Bas et al., 2016, Spain [24]	Quantitative cross-sectional	Self-report questionnaires (TMMS-24, LOT-R, etc.)	Analyze the influence of perceived emotional intelligence, optimism, and rumination on dropout risk in nursing students	285—third-year nursing students; mean age—not reported	BSN University	Dropout	Emotional intelligence Optimism Rumination
Sharif-Nia et al., 2023, Iran [25]	Cross-sectional study	Self-administered online questionnaire	Explore the relationships between experiences of bullying and intentions to drop out among Iranian nursing students, with major satisfaction and a sense of belonging serving as mediating factors.	386 undergraduate nursing students— 22.63 years	Undergraduate nursing programs at Alborz and Mazandaran Medical Sciences Universities	Bullying behaviors, Sense of belonging, Dropout intention	Bullying and students’ intentions to drop out
Soerensen et al., 2023, Denmark [32]	Exploratory, qualitative	Telephone interviews	Explore the students’ experiences leading to dropping out to gain a deeper understanding of their perspectives	15 students—ages 21–32	University College	Dropout	Reasons for dropout
Ten Hoeve et al., 2017, The Netherlands [28]	Exploratory, qualitative	Semi-structured telephone interview	Examine the factors that affect student nurses’ decisions to leave or complete their program	17 students—ages 19–33	Four Universities of Applied Sciences	Dropout	Factors affecting student nurses’ decision to leave or complete their program
Van Hoek et al., 2019, Belgium [26]	Cross-sectional design	Survey	Explore the influence of socio-demographic factors, resilience, and stress-reducing activities on academic outcomes among undergraduate nursing students	554 students—27.0 years	Six nursing colleges	Dropout Intention to leave	Intention to leave Academic success Dropout
Viottini et al., 2024, Italy [34]	Pilot multimethod study	Baseline quantitative online survey and follow-up semi-structured qualitative interviews	Understand the relationship between motivations for enrolment and dropout among first-year undergraduate nursing students	759 students - median age-20, 31 students were interviewed	Five Italian universities offering Bachelor of Science in Nursing programs	Dropout	Main reasons for dropout
Williams, 2010, USA [36]	Qualitative	Interview	Describe common experiences and practices that helped students persist and flourish during the first part of BNP	10 students—Mean age—not reported	College of Nursing	Attrition Persistence	Reasons affecting attrition
Wray et al., 2017, UK [20]	Quantitative, retrospective	The institution’s student record system	Map student characteristics at entry to the program against third-year completion data to examine non-progression and successful progression	725 students—Mean age—not reported	Nursing school	Drop out	Successful completion Non-successful completion: Academic reasons Unsuccessful completion: non-academic reasons

**Table 3 nursrep-15-00182-t003:** Factors associated with attrition among bachelor’s nursing students.

Author/Year/Country	Academic Factors	Institutional and Social Factors	Personal Factors	Economic/ Financial Factors
Abele et al., 2013, USA [19]	High attrition rates resulting from academic probation and course failures Poor performance in specific courses, such as psychology and microbiology	Not reported	Not reported	Not reported
Ashghali Farahani et al., 2017, Iran [29]	Obligation to choose nursing (cultural and legal circumstances that compelled participants to choose nursing). Lack of preparation before clinical practice. Heavy academic workload (numerous, time-consuming assignments). Academic atmosphere. Insufficient management	Improper teacher-student ratio Discrepancy between expectations and actual experiences Poor workforce management and inadequate supervision (including failure to maintain an appropriate student-teacher ratio) Shared education (nursing students feel subordinate to physicians and unable to contribute their scientific or practical knowledge in clinical settings) Negative influence from practicing nurses Low social prestige (lack of professional identity or societal recognition)	Male student (Embarrassment of working as a nurse) Lack of personal, professional identity Abuse from nurses in clinical practice	Not reported
Bakker et al., 2021, The Netherlands [21]	High psychological demands Supervisor and co-worker support. Psychological distress	Lack of institutional support during clinical placements Experiences of discrimination and poor social integration	Not reported	Not reported
Barbé et al., 2018, USA [35]	Lower confidence in study skills (note review, exam prep) Lower ability to complete reading load.	Born to immigrant families. Born outside the U.S English not spoken at home. Perceived discrimination Racial/ethnic minority status	Lower self-esteem Feelings of inadequacy Less belief in academic ability	Financial concerns Inability to purchase textbooks and required electronics. No direct association with tuition/aid
Canzan et al., 2022, Italy [3]	Struggles with academic workload Lack of organization and study skills Exam preparation challenges	Disparity between the ideal of nursing and the reality experienced during clinical placement Dissatisfaction with the overall clinical placement experience Perceived lack of support from the clinical instructor	Not being suited for nursing. Perception of lacking the psychological, physical and practical resources needed to cope with nursing school and profession Anxiety, emotional burden. Low motivation Mismatch between expectations and reality	Not reported
Dancot et al., 2021, Belgium [33]	No direct academic factor specified	Not reported	Low self-esteem (low self-liking and self-competence)	Not reported
Kox et al., 2022, The Netherlands [22]	Absence due to illness during the academic year.	Living situation (not residing with parents) Limited decision latitude (students with fewer opportunities to make independent work-related decisions are more likely to drop out) Support from peers and colleagues	Gender (Male sex)	Not reported
Kukkonen et al., 2016, Finland [27]	Satisfaction with the program Lack of studying skills Practical orientation	Negative clinical experiences No realistic job view or perception of nursing as a profession Lack of support in transitioning from high school to university Nursing does not meet students’ expectations	Wrong career choice Multiple roles (parenting, working, studying) Difficulties in combining study with one’s life situation (sickness or death of a relative) Unrealistic expectations	Not reported
Matteau et al., 2023, Canada [23]	Effort-reward imbalance High academic demands (efforts)	Limited support from faculty and mentors School-work–life conflict	High efforts and school-work–life conflicts Experiencing high psychological distress	Not reported
Mazzotta et al., 2024, Italy [30]	Poor academic preparation from high school. Insufficient study habits and skills Lack of clarity about the demands of the nursing program	Poor organization of courses and clinical placements Limited support during clinical placements. Inadequate awareness of academic support services	Unclear professional identity Emotional stress and anxiety Family-related issues	Financial obligations including tuition, transportation, living costs. Family responsibilities such as childcare or care for ill relatives
Roos et al., 2016, South Africa [31]	Academic non-performance	Clinical environmental difficulties Difficulty coping with university demands and clinical expectations	Illness and poor health. Personal problems. Wrong career choice	Financial reasons (Lack of financial aid)
Roso-Bas et al., 2016, Spain [24]	Not directly reported	Not directly reported	Pessimism Low emotional clarity/repair Depressive rumination	Not reported
Sharif-Nia et al., 2023, Iran [25]	Lower academic engagement	Bullying behaviors (verbal abuse, intimidation, exclusion) from faculty, classmates, and clinical instructors Negative impact of bullying on academic engagement, self-esteem, and sense of belonging	Psychological distress from bullying	Not reported
Soerensen et al., 2023, Denmark [32]	Feeling unprepared for challenges in clinical practice High academic workload Lack of feedback from clinical supervisors	Lack of personal and professional support in completing studies Lack of social well-being Feelings of loneliness Lacking a “sense of belonging” Social environment	Emotional vulnerability Personal experiences (e.g., illness, pregnancy) Lack of resilience	Not reported
Ten Hoeve et al., 2017, The Netherlands [28]	Problems with the training program Discomfort working in groups Theory–practice gap Insufficient practical skills training	Perceived lack of support from mentors and clinical team Not feeling welcomed	Personal circumstances (problems achieving learning goals, problems working in a team, uncertainty about own knowledge and abilities)	Not reported
Van Hoek et al., 2019, Belgium [26]	Not reported	Studying in a densely populated city	Lower resilience More destructive and less positive stress-reducing activities History of suicide attempt(s)	Not reported
Viottini et al., 2024, Italy [34]	Excessive academic workload Inadequate preparation for academic and clinical placement demands	Negative social image of nursing Limited career progression opportunities Negative experiences in clinical placements	Lack of interest in the nursing profession Physical demands of the profession Family, health problems, or the impossibility of balancing university and work commitments	Financial difficulties
Williams, 2010, USA [36]	Heavy course load during the early phase of the nursing program Difficulties with time management High academic expectations	Little or no attempt to establish bonding between the students Lack of student-teacher or peer connection.	Lack of time management skills and use of resources Academic success (poor academic performance) Lack of emotional support from family	Financial support from family
Wray et al., 2017, UK [20]	Students with a higher-level entry qualification	Not reported	Higher age on entry Domicile (e.g., living situation or distance from university)	Not reported

**Table 4 nursrep-15-00182-t004:** Academic and non-academic strategies suggested for retention.

Author Name/Year/Country	Academic Retention Strategies	Non-Academic Retention Strategies
Abele et al., 2013, USA [19]	Offering students the tutoring and support to succeed in the program such as additional courses designed to enhance students’ critical thinking abilities Identifying students at risk for academic failure and providing them with additional assistance prior to beginning the curriculum Meeting with the student periodically throughout the semester to provide resources and activities to help the student improve in the necessary competency areas Arranging student to student mentoring	Not reported
Ashghali Farahani et al., 2017, Iran [29]	Close supervision of both clinical and educational activities Providing more resources within the educational environment	Promoting awareness about the identity of nursing as a profession Efficient management of workforce provision Promoting professional sociability
Bakker et al., 2021, The Netherlands [21]	Co-worker support was identified as a protective factor for reducing dropout intentions Improving institutional support for students in clinical placements Addressing workplace violence to create a safer learning environment	Not reported
Barbé et al., 2018, The USA [35]	Individual and group tutoring (faculty-guided and peer-to-peer tutoring) Supportive networks of faculty, registered nurses, and peers from diverse backgrounds Proactive strategies to support student success, especially those targeted at diverse student populations Early intervention by nursing faculty and academic support staff to help students build confidence in their study skills Encouraging mentorship and a sense of belonging among minority and international students	Offering language support programs focused on English pronunciation, vocabulary building Offering tutoring to improve listening and note-taking skills, and verbal and nonverbal communication through role-playing scenarios Assessing students who lack access to resources and identifying creative, cost-effective ways to make resources accessible to disadvantaged students Addressing perceived discrimination and cultural mismatch
Canzan et al., 2022, Italy [3]	Supportive mentorships Intervention of peer leaders Creation of summer schools for future first-year students Tutorship in clinical training	Encourage potential nurses and midwives to reflect on the values, attitudes, and capabilities they need to succeed The creation of open day/week events targeting high school students, where students have the chance to attend nursing classes for several days and better understand what the main components of undergraduate nursing programs are
Dancot et al., 2021, Belgium [33]	Not explicitly reported	Address low self-esteem through confidence-building activities and mentoring
Kukkonen et al., 2016, Finland [27]	Better orientation to the academic nature of the program, especially for younger students Guidance to improve study skills	More flexible study arrangements during personal crises Improved mental health support. Early career guidance to ensure realistic expectations
Matteau et al., 2023, Canada [23]	Implementing structured support systems to help students manage workload and academic pressures Mental health programs to address psychological distress and overcommitment Balance between academic efforts and rewards to reduce effort-reward imbalance Institutional policies to create a more flexible academic structure Reducing school-work–life conflicts	Improve support systems to address school-work–life balance Promote social support Enhance reward structures
Mazzotta et al., 2024, Italy [30]	Enhance academic orientation at the beginning of the nursing program Improve clarity around nursing role expectations and academic requirements Strengthen organization of courses and clinical placements Offer targeted academic support (e.g., tutoring, skills development)	Provide emotional and psychological support to reduce stress and anxiety Facilitate financial assistance or economic support for students Improve faculty-student relationships and mentorship during clinical practice Foster social integration and professional identity through peer and faculty support
Roos et al., 2016, South Africa [31]	Academic assistance/clinical support	Financial assistance and wellness programs Provide career guidance and personal support systems
Roso-Bas et al., 2016, Spain [24]	Not reported	Develop emotional intelligence (especially clarity and repair) Promote optimism Provide psychological support services to reduce pessimism and rumination
Sharif-Nia et al., 2023, Iran [25]	Implement anti-bullying interventions targeting faculty and clinical instructors Support academic engagement initiatives	Foster a sense of belonging by enhancing peer relationships and support networks among students Develop interventions to prevent and respond to bullying
Soerensen et al., 2023, Denmark [32]	Educator involvement in guiding students through emotionally challenging learning situations Promoting learning environments that connect vulnerability with professional growth Providing structured feedback, academic guidance, and better preparation for clinical placements	More targeted efforts to improve the social environment in nursing education Fostering a sense of belonging to the nursing profession
Ten Hoeve et al., 2017, The Netherlands [28]	Strengthen cooperation between lecturers and mentors to ensure consistency and support throughout training Provide clear guidance and feedback from educators during both theoretical and clinical components of the program Reduce writing barriers Improve clarity of training expectations	Support students’ intrinsic motivation and career goals through meaningful engagement with the profession and role models Create a positive clinical environment where students feel welcomed, supported, and part of the team
Van Hoek et al., 2019, Belgium [26]	Support resilience-building among nursing students Provide academic interventions for at-risk students	Offer mental health support, particularly for students with a history of suicidal behavior Ensure adequate financial aid and social support, especially for students in urban settings
Viottini et al., 2024, Italy [34]	Improve preparation for study load and clinical demands Offer emotional and psychological support programs Provide flexible learning options for students with personal or work commitments	Enhance the public image of nursing through awareness campaigns Offer emotional and psychological support Increase flexibility for work-study balance
Williams, 2010, USA [36]	Time management skills Faculty support in building connections among students, their peers, and families Encouraging students to build a career path	Using available resources Involve families early (e.g., newsletters, introductory meetings) Foster community and belonging through student organizations and cross-level mentoring
Wray et al., 2017, UK [20]	Identify and support students with lower entry qualifications early Tailor academic support for younger students	Recruiting/attracting older, local students

## Data Availability

No new data were created or analyzed in this study. Data sharing is not applicable to this article.

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
