# Peer review of "Examining Factors Associated with Attrition, Strategies for Retention Among Undergraduate Nursing Students, and Identified Research Gaps: A Scoping Review"

_nursrep, 2025, doi:10.3390/nursrep15060182_

Round 1

Reviewer 1 Report

Comments and Suggestions for Authors

Dear authors,

Thank you for your work. As a university professor of nursing, I found it very striking.
I was surprised that, in the end, the most important academic and non-academic factor is social support. Although it is true that on certain occasions, due to the large number of students involved, it is difficult to dedicate sufficient time to each student.
I would like to ask if you consider that, in the end, another factor influencing the dropout rate not only of nursing students, but also of healthcare professionals, is the working conditions they face. Not only during their internships but also throughout their professional lives.
Finally, I consider it appropriate to update the search since it was carried out in 2024.

Kind regards

Author Response

Dear reviewer, please see responses below and highlights in the revised manuscript.

Comments 1: Thank you for your work. As a university professor of nursing, I found it very striking. I was surprised that, in the end, the most important academic and non-academic factor is social support. Although it is true that on certain occasions, due to the large number of students involved, it is difficult to dedicate sufficient time to each student.

I would like to ask if you consider that, in the end, another factor influencing the dropout rate not only of nursing students, but also of healthcare professionals, is the working conditions they face. Not only during their internships but also throughout their professional lives.

Response 1: Thank you for your thoughtful comment. We agree that nurses’ working conditions which students` observe during clinical placements play a significant role in shaping nursing students’ experiences and decisions. These early experiences often reflect the broader realities of nursing work and may contribute to dropout if students feel the profession does not align with their expectations or values. We appreciate you raising this point and have added a sentence in the conclusion to acknowledge this connection more explicitly.

Page 29, third paragraph, line 4

Comments 2: Finally, I consider it appropriate to update the search since it was carried out in 2024

Response 2: Thank you for your valuable suggestion. We appreciate your attention to the timeliness of the literature search. In response, we confirm that the database search was initially conducted in May 2023 and subsequently updated in February, 2025 to ensure that the review reflects the most current and relevant studies published up to the end of 2024. This update is clearly reported in both the Abstract and the Methods section (Section 2.2.1, Data Sources and Search Strategy), and the PRISMA flow diagram has been revised accordingly to reflect the complete and updated search process.

Page one, abstract, Line 10

Page 5, First paragraph, line 1

 Best regards

Reviewer 2 Report

Comments and Suggestions for Authors

Interesting topic on a timely topic.

The citations that 50% attrition of first year students in the United States leave was not current. When checking the references, one is dated 2015 and the other 2022 however the data was collected in 2018-2019. If these are truly the latest studies in this area, please state that.

What specific inclusion criteria was used as you filtered the articles you found? I know you mentioned in English, etc but you went from 1416 retrieved to 19 - there needs to be more detail.

Author Response

Dear reviewer, 

please see responses below and the highlights in the manuscript.

Comments 1: The citations that 50% attrition of first year students in the United States leave was not current. When checking the references, one is dated 2015 and the other 2022 however the data was collected in 2018-2019. If these are truly the latest studies in this area, please state that.

Response 1: Thank you for the valuable comment. We have updated the nursing students` attrition rate in the United States and revised the manuscript with relevant references.

(Page 1, last paragraph and Page 2 First Paragraph)

Comments 2: What specific inclusion criteria was used as you filtered the articles you found? I know you mentioned in English, etc but you went from 1416 retrieved to 19 - there needs to be more detail.

Response 2: Thank you for your valuable feedback. We agree that more detailed clarification was needed regarding how we filtered the initial 1,416 records down to the final 19 included studies. We have now expanded the Methods section to clearly describe the inclusion and exclusion criteria, and we provided a PRISMA flow explaining the  screening process with updated search.

Page 4, first and second paragraphs

Best regards

Reviewer 3 Report

Comments and Suggestions for Authors

The overall quality of the article is high, but there are some methodological and editorial limitations that should be addressed before publication. Positive aspects, areas needing improvement, and specific suggestions for changes are detailed below:

As is common in reviews, it is important to conduct a formal quality assessment of the included studies. In its absence, this could weaken the reliability of the findings. It is recommended to include at least a brief critical appraisal of the selected studies. A critical appraisal of the included studies using validated tools, such as the Critical Appraisal Skills Program (CASP), is recommended to strengthen the reliability of the findings.

Most of the included studies are from specific contexts (e.g., European countries), and this limits the applicability of the findings to other cultural and educational contexts. Further discussion of how the findings can be applied to different cultural and educational contexts, especially in countries with limited resources, is recommended.

Some sections of the text are repetitive, especially in the discussion of strategies, and this may reduce the flow of the article. On the other hand, there are inconsistencies in the wording of certain terms (e.g., "attrition" and "dropout" are used interchangeably without a clear definition of what each means or whether they are the same). It is recommended to adopt consistent terminology. Some sentences have grammatical errors or ambiguities that should be corrected to improve the clarity and professionalism of the document. A revision of the text is recommended to eliminate repetitions and improve clarity. Adopt uniform terminology (e.g., define "attrition" and use it uniformly) and correct grammatical errors to ensure that the style is consistent with journal standards.

Although gaps in the literature are mentioned, there is insufficient depth in how these could be addressed in future research. For example, a conceptual framework or theoretical model could be proposed to guide future studies. It is recommended that a conceptual framework or theoretical model be proposed to guide future research on nursing student retention.

Tables or graphs that visually summarize key findings, which could facilitate interpretation of the results, are not included. It is recommended that tables or graphs summarizing attrition factors and retention strategies, as well as gaps identified in the literature, be included.

Comments on the Quality of English Language

Some sentences have grammatical errors or ambiguities that should be corrected to improve the clarity and professionalism of the document. A revision of the text is recommended to eliminate repetitions and improve clarity. Adopt uniform terminology (e.g., define "attrition" and use it uniformly) and correct grammatical errors to ensure that the style is consistent with journal standards.

Author Response

Dear reviewer, 

please see responses below and the highlights in the manuscript. 

Comment 1: The overall quality of the article is high, but there are some methodological and editorial limitations that should be addressed before publication. Positive aspects, areas needing improvement, and specific suggestions for changes are detailed below:

Response 1: Thank you for your constructive and encouraging feedback. We are grateful for your recognition of the overall quality of our work and for the detailed suggestions you provided. We have carefully reviewed all the methodological and editorial points raised and have revised the manuscript accordingly to strengthen its clarity, transparency, and scientific rigor. Specific responses to each of your suggestions are provided below.

Comment 2: As is common in reviews, it is important to conduct a formal quality assessment of the included studies. In its absence, this could weaken the reliability of the findings. It is recommended to include at least a brief critical appraisal of the selected studies. A critical appraisal of the included studies using validated tools, such as the Critical Appraisal Skills Program (CASP), is recommended to strengthen the reliability of the findings.

Response 2: Thank you for the feedback. The necessary changes in the method and in the limitations, were made.

Page 6, Third Paragraph, line 2

Page 28, last paragraph, line 3

Comment 3: Most of the included studies are from specific contexts (e.g., European countries), and this limits the applicability of the findings to other cultural and educational contexts. Further discussion of how the findings can be applied to different cultural and educational contexts, especially in countries with limited resources, is recommended.

Response 3:

Thank you for this insightful comment. We agree that the concentration of studies in European and other high-income settings may limit the generalizability of the findings. In response, we have expanded the Discussion section to reflect on how key themes such as academic pressure, social support, and clinical experiences may manifest differently in low- and middle-income countries due to resource limitations, cultural expectations, and institutional capacity. We also emphasize the importance of contextualized retention strategies and call for future research in underrepresented settings. A sentence acknowledging this limitation has also been added to the Limitations section.

Page 28, third paragraph

Page 28, last paragraph. Line 10

Comment 4: Some sections of the text are repetitive, especially in the discussion of strategies, and this may reduce the flow of the article. On the other hand, there are inconsistencies in the wording of certain terms (e.g., "attrition" and "dropout" are used interchangeably without a clear definition of what each means or whether they are the same). It is recommended to adopt consistent terminology. Some sentences have grammatical errors or ambiguities that should be corrected to improve the clarity and professionalism of the document. A revision of the text is recommended to eliminate repetitions and improve clarity. Adopt uniform terminology (e.g., define "attrition" and use it uniformly) and correct grammatical errors to ensure that the style is consistent with journal standards.

Response 4:

Thank you for your thoughtful feedback. We’ve gone through the manuscript carefully and made several improvements based on your comments. First, we reduced repetition in the discussion of strategies by merging overlapping points—particularly those about academic support, mentoring, and faculty engagement—to improve the flow and avoid redundancy. We also addressed the inconsistent use of the terms “attrition” and “dropout.” I’ve now defined “attrition” clearly early in the manuscript and used it consistently throughout, except when referring to studies that specifically used the term “dropout.”

Lastly, we revised the text for grammar, clarity, and consistency. Several sentences were reworded to improve readability and ensure that the style meets the standards of the journal. We believe the manuscript is now much clearer and more polished as a result. The changes were highlighted in red throughout the manuscript.

Comment 5: Although gaps in the literature are mentioned, there is insufficient depth in how these could be addressed in future research. For example, a conceptual framework or theoretical model could be proposed to guide future studies. It is recommended that a conceptual framework or theoretical model be proposed to guide future research on nursing student retention.

Response 5:

Thank you for this insightful suggestion. We agree that the complexity of nursing student attrition requires a guiding conceptual approach for future studies. In response, we have incorporated a paragraph in the discussion introducing the “wicked problem” framework by Hamshire et al. (2019), which conceptualizes attrition as a complex and multi-system issue. We believe this framing provides a meaningful foundation for future research, emphasizing stakeholder engagement and system-level thinking.

Page 28, first paragraph

Comment 6: Tables or graphs that visually summarize key findings, which could facilitate interpretation of the results, are not included. It is recommended that tables or graphs summarizing attrition factors and retention strategies, as well as gaps identified in the literature, be included.

Response 6: Thank you for your thoughtful suggestion. We agree that visual summaries are helpful for interpretation. However, the manuscript already includes five comprehensive tables presenting attrition factors, retention strategies, and research gaps. We felt that adding additional tables or figures might lead to visual and structural overload, potentially reducing clarity for the reader. For this reason, we have chosen to retain the current structure, but we truly appreciate your suggestion and remain open to further guidance from the editorial team.

Comment 7: Comments on the Quality of English Language

Some sentences have grammatical errors or ambiguities that should be corrected to improve the clarity and professionalism of the document. A revision of the text is recommended to eliminate repetitions and improve clarity. Adopt uniform terminology (e.g., define "attrition" and use it uniformly) and correct grammatical errors to ensure that the style is consistent with journal standards.

Response 7: Thank you for your constructive feedback. In response, we carefully revised the manuscript to improve clarity, correct grammatical errors, and ensure consistent language throughout. We also addressed repetition by merging overlapping content and streamlining sections for better flow. To ensure terminological consistency, we defined the term “attrition” early in the manuscript and have used it uniformly throughout the text, replacing other terms such as “dropout” or “withdrawal” unless used in the context of cited studies. We believe these revisions enhance the professionalism and readability of the manuscript in line with journal standards.

Best regards,